# Impact of Cloud Condensation Nuclei Reduction on Cloud Characteristics and Solar Radiation during COVID-19 Lockdown 2020 in Moscow

**Julia Shuvalova** [1,2,*], **Natalia Chubarova** [1] and **Marina Shatunova** [1,2]

1   Department of Meteorology and Climatology, Faculty of Geography, Lomonosov Moscow State University, 119991 Moscow, Russia
2   Laboratory of Detailed Numerical Weather Forecasts, Hydrometeorological Research Center of the Russian Federation, 123242 Moscow, Russia
*   Correspondence: shuvalova@mecom.ru; Tel.: +7-499-255-1309

**Abstract:** We used MODIS observations to retrieve number concentration of cloud droplets ($N_d$) at cloud lower boundary during spring 2018–2020 for the Moscow region. Looking through the similar synoptic situations of the northern clear air advection, we obtained $N_d$ within the limits of 200–300 cm$^{-3}$. During the lockdown period, with similar northern advection conditions, the reduction of $N_d$ on 40–50 cm$^{-3}$ (or 14–16%), with the increase in droplet effective radius by $8 \pm 1\%$ and cloud optical thickness reduction by $5 \pm 2\%$, was observed in contrast to the values in typical conditions in 2018–2019. We used these values for setting up corresponding parameters in numerical simulations with the COSMO-Ru model. According to the numerical experiments, we showed that the observed reduction in cloud droplet concentration by 50 cm$^{-3}$ provides a 5–9 W/m$^2$ (or 9–11%) increase in global irradiance at ground in overcast cloud conditions with LWP = 200–400 g/m$^2$.

**Keywords:** cloud condensation nuclei; aerosol indirect effects; COSMO-Ru; MODIS; COVID-19 lockdown

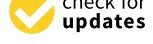



## 1. Introduction

Spring 2020 is remembered due to the unique situation of the COVID-19 pandemic lockdown [1]. Reductions in air pollution as an effect of activity restrictions were noted in China [2], India [3], the USA [4], and many European cities [5–7]. In Moscow, the lockdown period continued from the end of March until 9 June 2020. Restrictive measures superposed atmosphere circulation features and led to a significant reduction in emissions of pollutants, concentrations of small gas species and particulate matter (PM2.5, PM10) [8–10].

Variations in aerosol concentration, which can act as cloud condensation nuclei (CCN), have a significant impact on cloud microphysics, its optical and radiative properties and, hence, on the Earth energy budget [11–13], providing so called indirect or aerosol–cloud interaction (ACI) effects. However, still this phenomenon is characterized by large uncertainties [13]. Recent studies show an influence of aerosol content reduction during the lockdown period on cloud parameters: significant correlations between changes in PM2.5 concentrations and thunderstorm cloud activity [14,15], impacting on cloud lower boundary height [16]. The decrease in aerosol concentration over the Indo-Gangetic Plane may lead to a weakening of the Aerosol Invigoration Effect [17]. On the other hand, the effects due to a decrease in aerosol concentration were not revealed for China [18]. In addition, aircraft data and numerical simulations show insignificance of the aerosol indirect effect due to its high variability for Europe [19]. Thereby, the role of aerosol concentration reduction in aerosol–cloud interaction is defined by regional and meteorological conditions.

Relation between cloud droplet number concentration ($N_d$) and number concentration of cloud condensation nuclei ($N_{CCN}$) is widely used in retrieval methods and simulations.

Usually, $N_{CCN}$ is assumed equal to $N_d$ at cloud base, precisely within the first tens of meters above the cloud base. $N_{CCN}$ can also be related to aerosol optical thickness (AOT) [20] or hydrophilic aerosol concentration $N_{aer}$ [21]. However, this approach is criticized because of the incomplete correspondence of the recovered $N_d$ to the integral AOT values and because AOT and $N_{aer}$ do not fully represent aerosol physical properties. The uncertainty of $N_{CCN}$ retrievals increases for the unpolluted atmosphere when the optical effect of the aerosol is small [22].

Another solution is the application of the methods based on physical relationships of $N_d$ with other cloud characteristics, such as liquid water content or liquid water path, droplet size, cloud optical thickness and radar reflectivity [23–28].

In this paper, we estimated the possible effect of $N_d$ reduction during the spring lockdown in Moscow on cloud characteristics and global irradiance. $N_d$ retrieval methods based on satellite measurements are described in Section 2. The results for the lockdown period in comparison with the previous year's data are presented in Section 3.1. In Sections 3.2 and 3.3, we discuss an effect of variations in $N_d$ on cloud characteristics and radiation according to observations and simulations.

## 2. Materials and Methods

We evaluated the aerosol indirect effects based on observations and simulations of COSMO-Ru numerical weather prediction model. A brief flowchart of the research is shown in Figure 1. Further, the applied observation data, methods of $N_d$ retrievals, the structure of numerical experiments, analysis and comparisons of results are presented in detail.

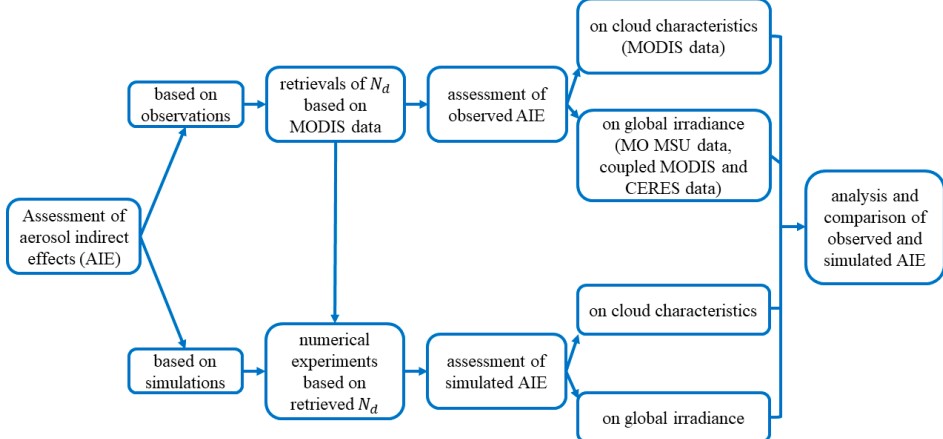

**Figure 1.** A flowchart of research aerosol indirect effects over Moscow using observations and numerical experiments.

We used MODIS dataset (Collection 6.1 Level 2) [29] to retrieve $N_d$ values, which are available at LAADS DAAC (https://ladsweb.modaps.eosdis.nasa.gov, accessed on 1 February 2021). The main information about the MODIS data used in this study is shown in Table 1. Cloud microphysical and optical parameters are calculated by MODIS bispectral algorithms using observations at wavelength within water absorption lines and out of them [30]. MODIS cloud water path is determined proportionally to the retrieved MODIS cloud optical thickness and MODIS effective radius of cloud droplets [31].

**Table 1.** A brief description of the MODIS data used in this study.

| Characteristic | Data Resolution, km | Method of Observation |
|---|---|---|
| Cloud optical thickness | 1 | [30] |
| Liquid water path | 1 | [31] |
| Droplets effective radius | 1 | [30] |
| Cloud multi-layer flag | 1 | [32] |
| Cloud phase infrared | 1 | [31,33] |
| Water vapor path | 1 | [34] |
| Cloud fraction | 5 | [31] |

We calculate $N_d$ values using the two following methods. The first one (hereafter, method 1) [35] defines the relationship of $N_d$ with liquid cloud optical thickness ($COT_{liq}$) and droplet effective radius ($R_{eff}$) as follows:

$$N_d = k_1 COT_{liq}^{0.5} R_{eff}^{-2.5}, \qquad (1)$$

where $k_1$ is equal to $1.37 \times 10^{-5}$ m$^{-0.5}$.

According to [27] (hereafter, method 2), $N_d$ is connected with liquid cloud optical thickness and liquid water path (LWP):

$$N_d = k_2 COT_{liq}^{3} LWP^{-2.5} \qquad (2)$$

where $k_2$ is a constant equal to $157.216$ kg$^{5/2}$m$^{-8}$.

The above expressions are valid for adiabatic liquid low-level cloud (cloud phase infrared and cloud multi-layer flag (see Table 1)), where the cloud droplet number concentration is assumed to be vertically constant and the liquid water content linearly depends on the vertical extent of the cloud and the coefficient of wet adiabatic condensation. The cloud condensation nuclei number concentration is assumed equal to the cloud droplet number concentration. A detailed description of this approach can be found in [36].

There are other more complicated approaches considered: inhomogeneity in solar radiation within the observation cell [37], cloud adiabaticity based on the analysis of satellite measurements at different wavelengths [38] and additional data filtering [22]. However, the comparisons with aircraft observations [39] showed that there was no significant accuracy improvement if these more sophisticated techniques were used.

Cloud features may depend not only on the presence of aerosols as condensation nuclei, but also on the synoptic processes [40,41], which can be described by specification of the air mass water vapor path. We used near-infrared (NIR) spectral MODIS data with 1 km horizontal resolution to evaluate water vapor path (see Table 1), since the observations in NIR spectrum have less air temperature sensitivity compared to infrared-range (IR) data [42]. Water vapor path was analyzed only for cloudless pixels (95% clear sky probability) [43].

We selected the days during springtime in 2018–2020, which corresponded to the restrictions of the applied methods described in [44]. Given the uncertainty in the retrieved $N_d$ increases in the presence of mixed and multilayer clouds, we chose the cases of one- and two-layer liquid clouds. To decrease observation errors of solar radiation at ground and at the top of the atmosphere we examined data when solar height was higher than 25° and radiometer zenith angle was less than 50°. Zinner and Mayer [45] showed that in the case of broken cloudiness, the underestimation of the retrieval cloud characteristics reaches 20%, while for a continuous overcast cloud cover error is about ±5%. The dependence of $N_d$ retrieval accuracy on cloud amount [46] limits the sample to situations with overcast cloudy conditions with cloud amount more than 90%.

Since MODIS cloud data have 5 km horizontal resolution, while the other data have 1 km resolution (see Table 1), we compared $N_d$ retrievals for both spatial resolution options. We

also used additional filtering, excluding data, when cloud optical thickness (COT) was less than 5 ($COT_{liq} < 5$) and effective droplet radius ($R_{eff}$) was less than 4 μm ($R_{eff} < 4$ μm) [47].

A significant effect of a change in the concentrations of pollutants over Moscow was noted not only for the entire lockdown period, but also for the periods of the different type air mass advections [8].

Here, we used periods with quasi-homogeneous meteorological conditions (QHMC) in April 2018–May 2020 in the Moscow region, which were defined [8,48] taking into account observations on atmospheric pressure, the predominant winds and day-to-day variations in meteorological parameters together with conditions for the intensity of pollution dispersion (IPD) [49]. Amid the QHMC periods, we chose ones with northern advection to eliminate overestimated lockdown effects and to reduce the influence of large-scale processes on the variability in $N_d$ [50].

We identified 4 days for numerical simulations among ones with northern advection when all above-mentioned conditions were satisfied: 22 April 2018, 31 May 2018, 8 and 22 March 2020. These days were characterized by the presence of stratiform or stratocumulus optically thick clouds over Moscow. This condition was applicable to ensure a better '$N_d$-$N_{CCN}$' agreement [51]. The days with precipitation [24], the presence of smoke aerosol and high wind speed (>10 m/s) were also excluded from the sample. Temperature conditions were typical for northern air advection within a range of 2.4–10 °C. Water vapor path (WVP) median values varied from 8.5 up to 12.9 kg/m² that corresponds to the Arctic air moisture content, taking into account the intra-annual variation and air mass transformation during advection [52,53].

The northern advection air masses coming to Moscow region originate from Northern Atlantic, Arctic Ocean, the northern parts of Siberia and European part of Russia. There are not any major emission sources of PM$_{2.5}$ and PM$_{10}$ in these areas according to EMEP (https://www.ceip.at/, accessed on 15 March 2021) and TNO-MACC II (https://eccad.aeris-data.fr, accessed on 15 March 2021). Usual annual anthropogenic emissions of PM$_{2.5}$ and PM$_{10}$ based on EMEP database (flux in μg/(m²s), year 2009) and TNO-MACC II database (emission in Tg per year, year 2017) are shown in Figure 2.

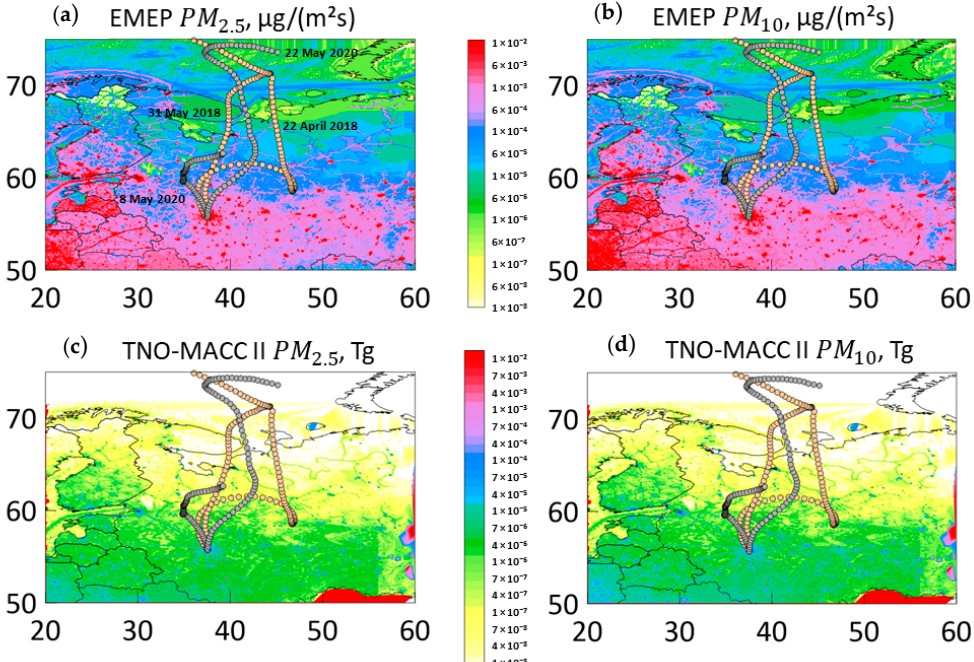

**Figure 2.** Annual anthropogenic emissions of PM$_{2.5}$ and PM$_{10}$ over Europe from EMEP database for 2009 ((**a**,**b**), in μg/(m²s)) and TNO-MACC II database for 2017 ((**c**,**d**), in Tg per year). The lines are the backward trajectories from HYSPLIT model at 500 m above ground for NWP cases (12 UTC of each day).

HYSPLIT (https://aeronet.gsfc.nasa.gov/, accessed on 15 March 2021) calculations (trajectories in Figure 2) showed that in the lower troposphere the trajectories did not pass through large industrial cities. Thus, remote sources of emissions did not affect the obtained $N_d$ differences between 2018–2019 and 2020.

As a result, we obtained three samples. The first sample contains all available data subject to limitations on satellite observations. The second sample (NA cases) was obtained from the first one by selecting cases of northern advection and the third sample (NWP cases) was obtained from the second one with the use of more stringent selection conditions.

A summary of the observed cloud parameters and retrieved $N_d$ for the three samples is presented in Table 2. NWP cases account for 15% of the second sample (NA cases) and 9% of the all cases. Cloudiness in NWP cases is characterized by larger values of liquid water path and optical thickness.

**Table 2.** Median and interquartile range (shown through a slash "/") of the observed and retrieved cloud characteristics for the samples. Errors were provided with the data by the developers.

| Characteristics | Data Grid Step, km | All Cases | Only Northern Advection Cases (NA Cases) | Selected Northern Advection Cases for Numerical Experiments (NWP Cases) |
|---|---|---|---|---|
| Number of days | | All: 116 | All: 66 | All: 4 |
| | | 2018: 26 | 2018: 11 | 2018: 22/04, 31/05 |
| | | 2019: 42 | 2019: 16 | |
| | | 2020: 48 | 2020: 39 | 2020: 08/05, 22/05 |
| Effective radius of cloud droplets ($R_{eff}$), μm | | 9/4 | 9/4 | 11/4 |
| $R_{eff}$ error, % | | 7/2 | 6/1 | 6/1 |
| Liquid water path (LWP), g/m$^2$ | 1 | 106/138 | 120/150 | 151/175 |
| LWP error, % | | 16/6 | 16/6 | 15/4 |
| Liquid cloud optical thickness ($COT_{liq}$) | | 19/21 | 21/22 | 23/22 |
| $COT_{liq}$ error, % | | 7/5 | 7/5 | 7/4 |
| Number of points | | 338065 | 187152 | 28809 |
| Number of points | 5 | 19705 | 11979 | 1371 |
| $N_d$ according to method 1, cm$^{-3}$ | | | | |
| 2018–2020 2018–2019 2020 | 1 | 246/274 252/270 232/281 | 250/286 272/292 232/278 | 188/159 240/188 144/114 |
| 2018–2020 2018–2019 2020 | 5 | 213/250 220/244 201/256 | 223/254 243/256 208/259 | 167/133 212/132 129/112 |
| $N_d$ according to method 2, cm$^{-3}$ | | | | |
| 2018–2020 2018–2019 2020 | 1 | 280/322 288/318 264/331 | 285/336 310/345 263/327 | 212/185 272/220 161/130 |
| 2018–2020 2018–2019 2020 | 5 | 259/309 267/303 245/321 | 272/327 300/331 251/321 | 194/153 254/169 151/123 |

The results of the standard MODIS bispectral algorithm were used with a base wavelength of 2.1 μm. Their measurement errors generally correspond to the estimates of the

observation uncertainties in the updated version of Collection 6. The uncertainties in the retrieved $R_{eff}$ are about 10% and for $COT_{liq}$ are up to 8% for the liquid clouds. Measurement uncertainties for each pixel provided by the developers are on average 6–7% for the droplet effective radius and 6–7% for the optical thickness of liquid clouds. These errors include instrument calibration, atmospheric corrections and errors in the atmospheric model used to reconstruct cloud characteristics [29].

We obtained global solar irradiance data at ground from satellite CERES retrievals (SSF FM2-FM3) [54] and from ground-based measurements from the Meteorological Observatory of the Moscow State University (MO MSU) (http://www.momsu.ru, accessed on 1 August 2022). The latter dataset included global irradiance data from Kipp&Zonen CNR4 and direct radiation data from M-3 actinometer. Kipp&Zonen CNR4 global irradiance measurement error is less than 5% (https://www.kippzonen.com, accessed on 1 August 2022) and the uncertainty in direct radiation observations by M-3 actinometer is less than 4%. Ground-based observations were averaged over one-hour interval.

CERES data were synchronized with 1 km resolution MODIS data within 20 min time window and within a radius of up to 1 km. In this work, we used CERES retrieved radiative flux data using Model B [55], which is based on Langley Parameterized Shortwave Algorithm. Studies of the shortwave downward radiation and the radiation balance at the Earth's surface under cloudy conditions showed a good agreement between the CERES data and ground-based measurements at different points on the planet [56,57]. The deviation in CERES shortwave radiation relative to observations at continental stations in cloudy conditions is on average 2–4% [54].

We carried out numerical experiments with the COSMO model with 1.1 km grid spacing [58] over the Moscow region for NWP cases. The region of simulations is shown in Figure 3.

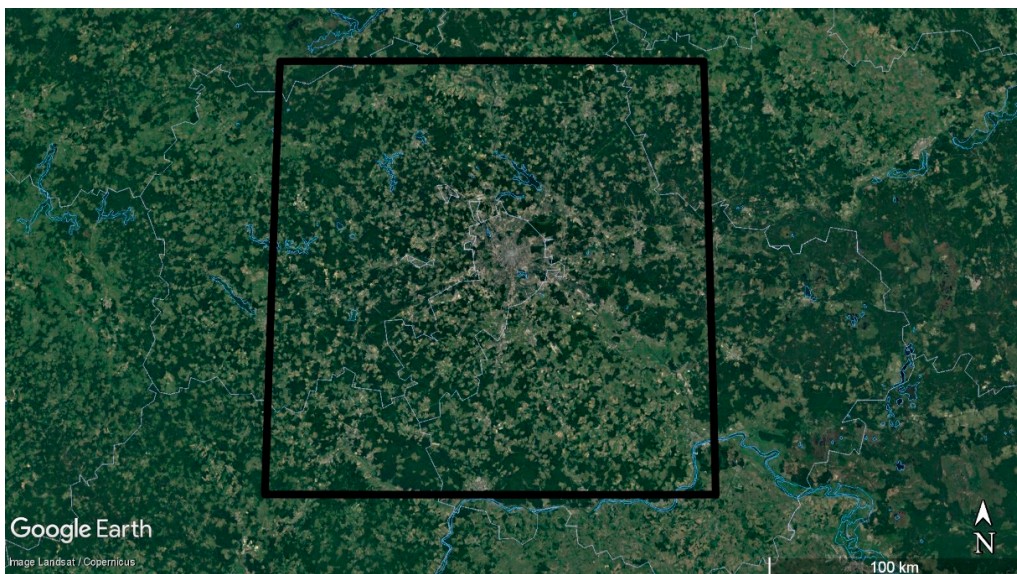

**Figure 3.** The area of numerical experiments. The borders of Russian regions are shown in white. The Moscow metropolis is located in the center. The figure was created using Google Earth Pro.

The simulations were performed with a 24 h lead time from 00 UTC. Two-moment microphysical scheme [59] was applied. In the COSMO model, radiative transfer scheme is represented by the delta two-stream method [60] with the CLOUDRAD cloud–radiative interaction scheme [61]. The CLOUDRAD scheme takes into account the indirect effects of aerosol through the concentration of cloud condensation nuclei. CLOUDRAD works with several options for initial data to obtain $N_{CCN}$. In this work, the $N_{CCN}$ was set constant, which made it possible to conduct experiments based on space-averaged observational data and to estimate sensitivity to this characteristic. The $N_{CCN}$ in the CLOUDRAD is assumed

equal to the number concentration of cloud droplets in the planetary boundary layer with the exponential decrease above. Thereby, the number concentration of cloud condensation nuclei was equal to the number concentration of cloud droplets both for measurement data and for numerical experiments.

## 3. Results

### 3.1. Number Concentrations of Cloud Droplets

First, we calculated $N_d$ by the two mentioned retrieval methods for all types of air masses over Moscow for the springs in 2018–2020 to obtain a general picture for the spring period (Figure 4a). Statistics of the retrieved data are presented in Table 2.

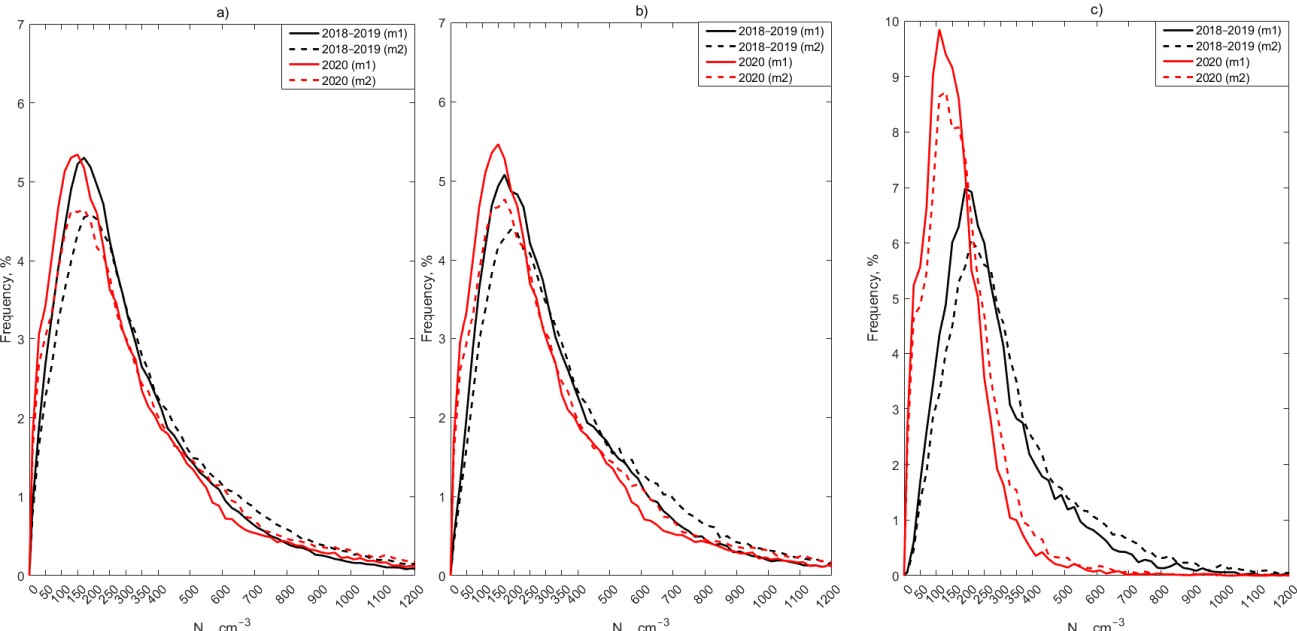

**Figure 4.** Frequency distribution of $N_d$ by the two methods in April–May 2020 and 2018–2019 for three samples: (**a**) all cases, (**b**) northern advection cases, (**c**) NWP cases.

On average, the lockdown spring of 2020 is not distinguished by particularly low concentrations compared to other years. This is because $N_d$ depends more on the synoptic conditions and on the prevalence of specific situations during the period. The median of daily $N_d$ was 184 cm$^{-3}$, 247 cm$^{-3}$ and 194 cm$^{-3}$ for 2018, 2019 and 2020, respectively. $N_d$ values in 2018 were the lowest among the presented years due to the colder spring conditions. The average spring air temperature at 2 m was 6 °C in 2018, 8 °C in 2019 and 7 °C in 2020. During the lockdown period in April–May 2020, during several periods, rather low air temperatures were also observed due to the predominance of northern air advection [8]. In addition, due to the lockdown itself in the Moscow region, the anthropogenic emissions from enterprises decreased by 20% [62]. Traffic load in the city and surrounding territories also decreased significantly, leading to a decrease in the concentration of aerosol precursor gases up to 50–60% [8]. According to [9], surface concentrations of particulate matter (PM$_{10}$) in the atmosphere of Moscow in April 2020 were 17% lower on highways and 28% lower in residential areas than the average for recent years. Road transport made a substantial contribution to the level of the final decrease in concentrations of suspended aerosol in the atmosphere [63]. A significant reduction in anthropogenic emissions in Moscow affected the content of potential condensation nuclei. In particular, during the northern advection, the average concentration of PM$_{10}$ during the lockdown period was 12 ± 0.4 μg/m$^3$, while before and after the lockdown, it was 17.3 ± 0.8 μg/m$^3$ [8].

In general, the $N_d$ range varied from several tens to thousands of particles per cubic centimeter. For all cases of spring periods in 2018–2020, $N_d$ median value was 246 cm$^{-3}$

according to method 1 and 280 cm$^{-3}$ according to method 2 (Table 2). These results are generally consistent with the results of other similar studies. The droplet concentration over the European part of Russia, on average, over the spring periods of 10 years, was about 300 cm$^{-3}$ according to MODIS data and about 200 cm$^{-3}$ according to CALIPSO data [64]. The N$_d$ range over Moscow also corresponds to the values typical for continental cloudiness. According to airborne measurements of continental clouds in the central regions of [65–67] the USA, N$_d$ was about 300–400 cm$^{-3}$ with maximum values up to 1000 cm$^{-3}$. The high variability in N$_d$ and the dependence on synoptic conditions are noticeable. This complicates the analysis of the N$_d$ effects, so we will further consider only the cases of northern advection. Figure 4b shows the frequency distribution densities of N$_d$ for the NA cases in the spring periods of 2018–2019 and 2020 according to these two methods. For 2018–2019, the median N$_d$ was 272 ± 3 cm$^{-3}$ according to method 1 and 310 ± 3 cm$^{-3}$ according to method 2, while in 2020, it was 232 ± 2 cm$^{-3}$ and 263 ± 3 cm$^{-3}$, respectively. Contrary to the sample with all cases, the sample with northern advection is characterized by a noticeable shift in the distribution to lower concentrations by 40–50 cm$^{-3}$ (14–16%) in 2020 relative to 2018–2019.

The N$_d$ distribution for the NWP cases (Figure 4c) is narrower and values higher than 300 cm$^{-3}$ are not presented practically for the days in 2020 compared to 2018–2019. The N$_d$ median values were 144 ± 2 cm$^{-3}$ (method 1) and 161 ± 3 cm$^{-3}$ (method 2) in 2020, while in 2018–2019 N$_d$ median values were 240 ± 4 cm$^{-3}$ (method 1) and 272 ± 4 cm$^{-3}$ (method 2), which is 80–110 cm$^{-3}$ (39–41%) larger than in 2020. The difference in retrieved N$_d$ between the two methods is about 14%, which can be attributed to the droplet effective radius influence [68].

Thus, the NA and NWP samples for 2020 are characterized by lower N$_d$ compared to those observed in 2018–2019, which can be explained by the influence of the lockdown on decreasing the level of air pollution.

Spatial distribution during the lockdown period is quite homogeneous. Contrarily, in 2018–2019, the N$_d$ values varied greatly and reached 2700 cm$^{-3}$ in some locations. The example of N$_d$ spatial distribution retrieved by method 2 for the days of the NWP sample is presented in Figure 5.

The median value of WVP and cloud characteristics are also shown below the maps for a particular day there. White areas on the maps indicate missing data. The low data availability on 8 May 2020 is associated with the presence of ice clouds over part of the territory, which prevented the N$_d$ retrieval over the entire region. Spatial variability in N$_d$ can be associated with the influence of the city itself. We considered the differences between N$_d$ values in the central part of the city and surrounding territory within the Moscow Ring Road. The excess of cloud droplet concentrations over the city center relatively to the surrounding area was small: on average, it was 9 ± 19 cm$^{-3}$ (5 ± 7%) according to method 1 and 19 ± 23 cm$^{-3}$ (9 ± 8%) according to method 2. The reasons for the relatively uniform field may be explained firstly by low level of the aerosol pollution, observed in Moscow, like in many other European cities [69,70] and by the absence of the sulfate hygroscopic aerosol emissions [62]. The vehicle emissions in the city have a major influence on condensation nuclei formation.

We also analyzed the data with coarser grid spacing of 5 km (Table 2). Decreasing the observational resolution leads to a decrease of the average N$_d$ values by 66 ± 13 cm$^{-3}$ (23%) according to method 1 and by 52 ± 12 cm$^{-3}$ (16%) according to method 2. The decrease in N$_d$ values is due to the increase in LWP values by 3% and R$_{eff}$ values by 10% along with a COT decrease by 3% in 5 km gridded data. However, using 5 km data with an additional condition for continuous cloud cover allows one to take into account subgrid inhomogeneity of cloudiness, which increases the horizontal component of radiative transfer and affects the retrieved cloud characteristics [47]. According to coarser resolution data, N$_d$ median values for NWP cases in 2018–2019 were 212 ± 12 cm$^{-3}$ and 254 ± 15 cm$^{-3}$, depending on retrieval method, while in 2020, they were 129 ± 8 cm$^{-3}$ and 151 ± 10 cm$^{-3}$, respectively

(Table 2). Thus, taking into account the cloud amount, $N_d$ median value is slightly reduced; however, the tendency towards the $N_d$ decrease during the lockdown period persists.

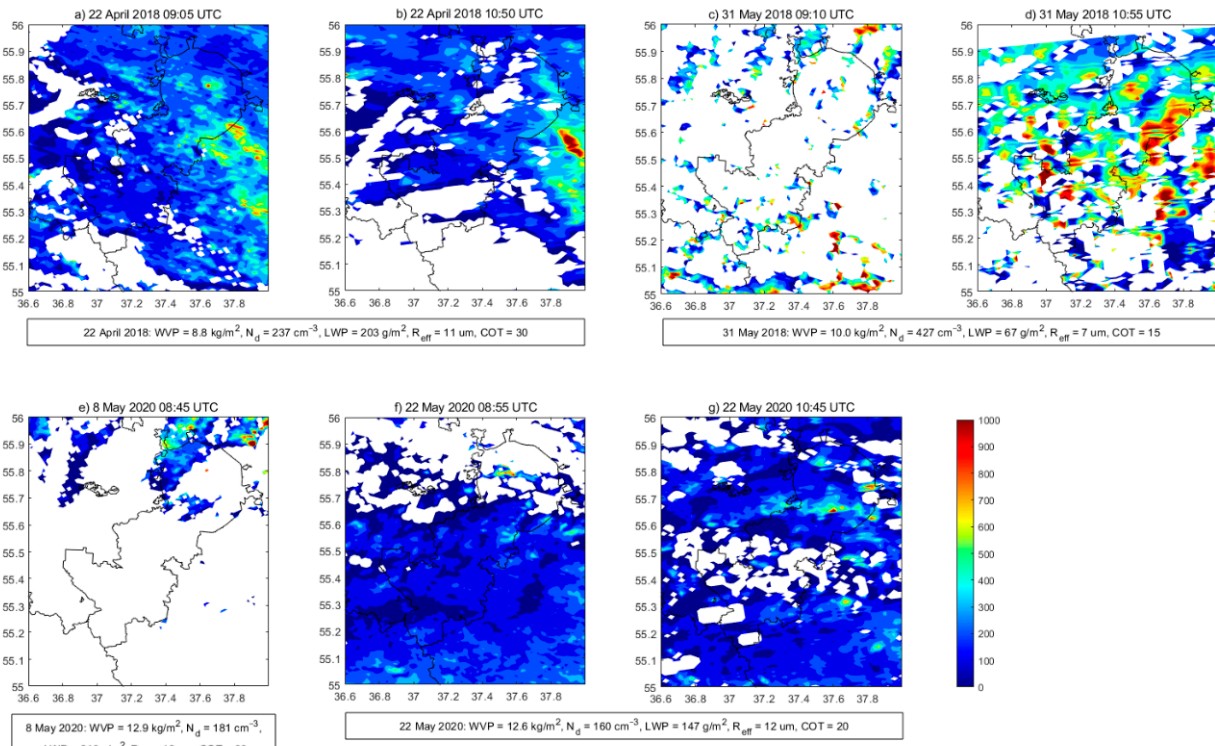

**Figure 5.** Spatial distribution of the droplet number concentration retrieved by method 2 for NWP sample. Each subfigure (**a**–**g**) describes an observational case.

Based on the retrieved values of $N_d$ from the MODIS data, for our numerical experiments, we chose $N_d$ equal to 100, 150, 200, 250 and 300 cm$^{-3}$, since these values represent typical cloud droplet number concentration over the Moscow region for the cases of northern air mass advection during the lockdown period and before it.

*3.2. Observed and Simulated First Aerosol Indirect Effect*

The changes in the cloud droplet number concentration were accompanied by the changes in other cloud parameters as well as by variations in global shortwave irradiance. In these experiments, we deal with the first indirect aerosol effect in observations and simulations; however, this effect may provide further changes in cloud properties. Correlations of liquid water path with droplet effective radius and liquid cloud optical thickness are presented in Figure 6a,b for two samples—NA and NWP—correspondingly for 2020 (shown in red) and 2018–2019 (shown in blue). LWP data are according to the MODIS observations with 1 km grid spacing. We used the intervals of liquid water path according to the evaluation of cloud–aerosol observations for 10 years, described in [71].

According to the MODIS retrievals, the decrease in cloud droplet number concentration during the lockdown period in 2020, together with the growth of droplet effective radius, leads to the COT reduction compared to the values observed in 2018–2019 (averaged difference is shown in plots). For the NA cases, this reduction is about $1.3 \pm 1.0$ ($5 \pm 2\%$) due to the $N_d$ decrease by $43 \pm 28$ cm$^{-3}$ ($12 \pm 7\%$) and the $R_{eff}$ increase by $0.8 \pm 0.1$ μm ($8 \pm 1\%$). For the NWP cases, the changes are larger: the COT decreases by $4.0 \pm 1.9$ ($18 \pm 4\%$), the $N_d$ decreases by $132 \pm 42$ ($45 \pm 9\%$) cm$^{-3}$ and $R_{eff}$ increases by $2.3 \pm 0.4$ μm ($25 \pm 5\%$).

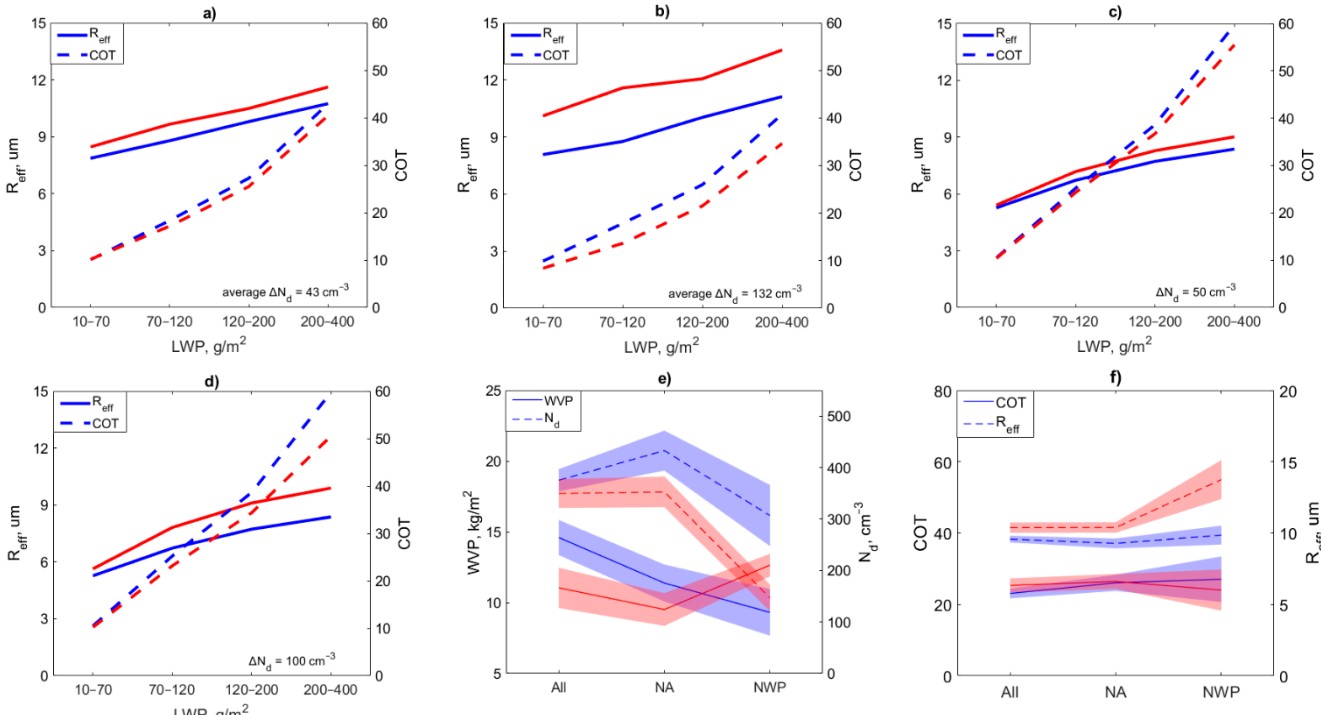

**Figure 6.** Lockdown 2020 effect on cloud characteristics for different samples: MODIS droplet effective radius and cloud optical thickness for NA (**a**) and NWP (**b**) samples; simulated maximum effective radius in the cloud profile and cloud optical thickness for NA (**c**) and NWP (**d**) samples; reduced samples (coupled MODIS and CERES data) for water vapor path and droplet number concentration (**e**); droplet effective radius and cloud optical thickness (**f**). Data for 2020 are marked by red, for 2018–2019, by blue colors.

The same relationships but according to the simulation are presented in Figure 6c,d. We set the $N_d$ difference of 50 or 100 $cm^{-3}$, thus, simulating the changes due to lockdown observed in the NA and NWP cases and taking the maximum simulated $R_{eff}$ along the vertical profile. The latter setting provides model $R_{eff}$ values closer to the satellite data estimates, where the measured effective radius corresponds to the top of the cloud. The maximum value also better reflects the effect of change in $N_{CCN}$ due to the peculiarities of the CLOUDRAD scheme, when using space-constant $N_{CCN}$ [61]. In Table 3, we summarize the observed and simulated features of the obtained results.

The indirect effects estimated in our work are comparable with the results obtained in other studies. Christensen [72] showed a COT increase by 28% and a decrease in the effective radius by 10% under conditions of polluted clouds. According to research by Liu [73] for China, a decrease in the Aerosol Index is accompanied by an increase in $R_{eff}$ by 20–30% and a decrease in COT by 25%.

**Table 3.** Observed and simulated changes in cloud microphysical and optical characteristics in spring 2020 compared to the values in springs 2018–2019.

| Data | Sample | $\Delta N_d$, $cm^{-3}$ | $\Delta R_{eff}$, µm | $\Delta COT$ | Number of Points |
|---|---|---|---|---|---|
| MODIS | NA | $-43 \pm 28$ ($-12 \pm 7\%$) | $0.8 \pm 0.1$ ($8 \pm 1\%$) | $-1.3 \pm 1.0$ ($-5 \pm 2\%$) | 84,866 |
| | NWP | $-132 \pm 42$ ($-45 \pm 9\%$) | $2.3 \pm 0.4$ ($25 \pm 5\%$) | $-4.0 \pm 1.9$ ($18 \pm 4\%$) | 15,068 |
| COSMO-Ru | NWP (variant 1) | $-50$ | $0.5 \pm 0.2$ ($6 \pm 2\%$) | $-1.8 \pm 1.7$ ($-4 \pm 2\%$) | 2,668,076 |
| | NWP (variant 2) | $-100$ | $1.1 \pm 0.5$ ($15 \pm 5\%$) | $-4.0 \pm 3.7$ ($-9 \pm 5\%$) | 2,684,304 |

The effects of lockdown on cloud characteristics over the Moscow region are significant for all cases of northern advection. They are not very large but quite noticeable. Based on satellite data for the same type of synoptic conditions of northern advection, we obtained a slight (about 14–16%) decrease in the $N_d$ in the spring of 2020 compared to $N_d$ in 2018–2019 with the increase in droplet effective radius by $8 \pm 1$% and cloud optical thickness reduction by $5 \pm 2$% (see Table 3). As mentioned above, this may be attributed to the decrease in industrial anthropogenic emissions up to 20% [62], as well as to the significant decrease in transport emissions, which, in turn, affected the concentration of aerosol and precursor gases [8,9]. The structure of anthropogenic emissions changed during the lockdown period. As shown in many studies for Moscow and other cities around the world, a decrease in transport activity is a main factor in emission reduction [74,75]. It can be assumed that a less pronounced decrease in the $N_d$ compared with the decrease in surface $PM_{10}$ concentrations of about 30–40% [8,74] can be explained by the location of the major-changed aerosol sources of emission near the ground and characteristics of planetary boundary layer over Moscow. The study of [76] noticed the complex structure of aerosol profile and high dependence of aerosol on the structure of the planetary boundary layer.

### 3.3. Global Irradiance at Ground

For evaluating the radiative effects of cloud–aerosol interaction and the effects of the lockdown conditions, we compared the CERES data with ground-based observations at the MO MSU. The measurements at the MO MSU were averaged over a 15 min window relative to the time of the satellite observations. All synchronous observations (73 cases in total) for April–May of 2018–2020 are presented in Figure 7a. Color indicates the distance between mean satellite point and ground-based point over the MO MSU. One can see that the CERES data are generally consistent with ground-based measurements, with a correlation coefficient of 0.84. The total CERES global irradiance is lower than the observations at the MO MSU by only $11 \pm 27$ W/m$^2$ (2%). However, the difference between satellite and ground-based observations slightly increases with the distance between the points. Thus, at a distance of up to 3 km, the standard deviation of the differences is 87 W/m$^2$, and at a distance of up to 5 km, it increases to 119 W/m$^2$. Thus, the CERES data within 0–2 km distance are in good agreement with ground-based measurements and can be used in further analysis.

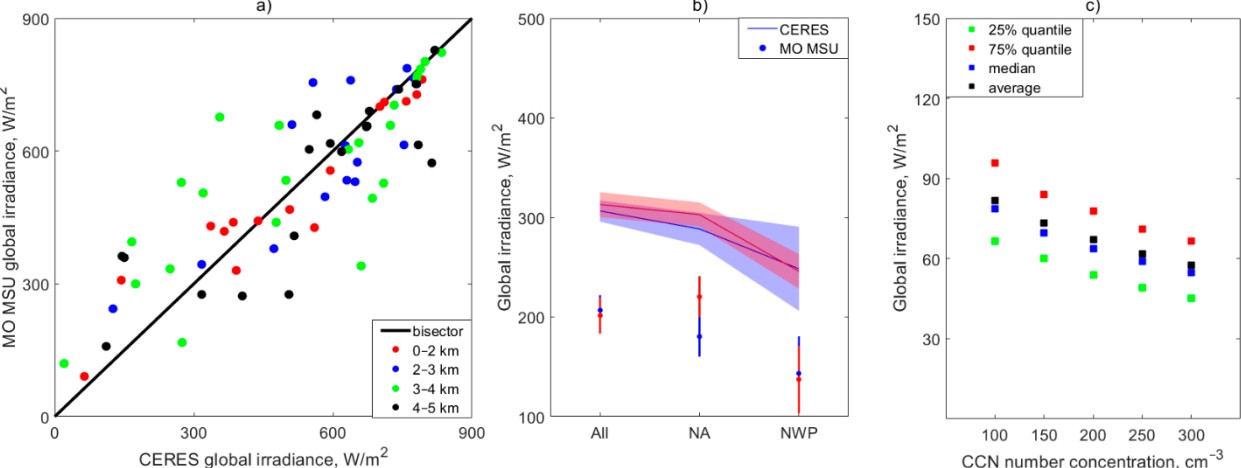

**Figure 7.** Observed and simulated global irradiance at ground: (**a**) CERES and MO MSU synchronous data (color indicates the distance between observations points); (**b**) CERES data over Moscow region (the same samples as for Figure 6e–f) and ground-based MO MSU global irradiance for three samples (data for 2020 is marked by red, for 2018–2019—by blue colors); (**c**) simulated global irradiance in dependence on $N_{ccn}$ for LWP = 200–400 gm$^{-2}$ for overcast condition. Solar height is equal 40°.

In order to analyze the joint cloud and radiative characteristics, we combined the MODIS measurements with the CERES data in a 20 min time window for those pixels, which were located within 1 km distance. Taking into account the spatial resolution of the CERES data, the number of cases in the samples was significantly decreased (all cases—1428 pixels, NA—847, NWP—105).

Mean values with confidence intervals for $N_d$, $R_{eff}$ and COT and WVP of the reduced samples are presented in Figure 6e,f. For all the cases of 2018–2019, WVP was about $14.6 \pm 1.2$ kg/m$^2$, while in 2020, due to the predominant northern advection during the period, it was $11.0 \pm 1.4$ kg/m$^2$. For the similar synoptic conditions observed during the northern advection in 2018–2019 and in 2020, WVP was $11.4 \pm 1.3$ kg/m$^2$ (in 2018–2019) and $9.5 \pm 1.2$ kg/m$^2$ (in 2020). These values are consistent with data for the transformed Arctic air masses [77]. Note that WVP is used only as an indicator of NA advections and the WVP difference itself did not affect the liquid water path of the NA sample (not shown).

The growth of the effective droplet radius with a decrease in the droplet number concentration for the reduced samples was equal to 1 μm and 4 μm for NA and NWP cases, respectively, in spring 2020 compared to previous years (see Table 3). At the same time, LWP values for the NA and NWP samples in 2020 were, on average, higher by 0.015 kg/m$^2$ and 0.036 kg/m$^2$, respectively, compared to 2018–2019 (not shown). Taking into account the combination of these two factors, the optical thickness of the NA and NWP clouds of the cases in 2018–2019 and 2020 differ little (Figure 6f). For the NWP sample, the cloud optical thickness in 2020 is, on average, lower than that in 2018–2019. However, with a significant decrease in the number of cases, the variability in COT increased.

CERES global irradiance data were adjusted to a solar height of 40° to exclude the influence of its small differences in the samples (Figure 7b). On average, global irradiance was higher by 14 W/m$^2$ in 2020 compared to that in 2018–2019 for the NA sample. At the same time, the difference was negligible for the NWP sample, because the lower droplet concentrations effects were compensated by the higher values of WVP and LWP for these days in 2020, which demonstrated possible effects of warmer and more humid air mass advection for these particular cases (see Figure 6e). We considered global irradiance observations at the MO MSU for the overcast conditions (Figure 7b). Ground-based observations as well as CERES satellite data show an increase of about 20–40 W/m$^2$ in global irradiance in 2020 compared to the previous two years for the northern advection cases.

We also simulated the global irradiance by setting the $N_{CCN}$ in a range of 100–300 cm$^{-3}$ with a step of 50 cm$^{-3}$ for evaluating the sensitivity of solar global irradiance to CCN; whereas COSMO model underestimated LWP compare to observations [78,79], here, we consider the sensitivity of the global irradiance to the CCN changes rather than determine the absolute cloud–aerosol radiative effect. Average LWP underestimation relative to MODIS data was $168 \pm 2$ g/m$^2$ ($42 \pm 1$%) in our experiments with $N_{CCN}$ of 300 cm$^{-3}$. This led to an overestimation of the global radiation on $112 \pm 15$ W/m$^2$ ($38 \pm 5$%).

From the experimental results, we obtained that the LWP (main factor in COT and, hence, Q modification) decreases by 4–6% (3–6 g/m$^2$) with an $N_{CCN}$ decrease of 50–100 cm$^{-3}$. For evaluating the direct relations of the global irradiance with $N_{CCN}$, we simulated their dependence for overcast conditions and adjusted results to the Sun's height of 40°. We also averaged the simulated global irradiance over the liquid water path within the interval of LWP = 200—400 g/m$^2$ to avoid the influence of the LWP variation. Thus, we considered the clear effect of $N_{CCN}$ on global irradiance through the size of cloud droplets without influence of the solar height and liquid water path. One can see a quasi-linear dependence between global irradiance and CCN (Figure 7c) with a 5–9 W/m$^2$ (or 9–11%) decrease for every 50 cm$^{-3}$ CCN increase.

Evaluated here, the COSMO-Ru simulated aerosol–cloud interaction effect on the global irradiance at the ground corresponds to the assessment of cloud–radiation interaction scheme in the COSMO model. According to the idealized numerical experiments of warm stratus, the global irradiance increases by about 20% when the CCN decreases from 200 to 100 cm$^{-3}$ [61] (Figure 6b). We obtained the growth of global irradiance by 9% in similar

conditions. Taking into account the differences in physical settings in the model, the Sun's elevation and cloud geometrical depth results are comparable.

## 4. Discussion and Conclusions

We presented the analysis of cloud condensation nuclei (CCN) number concentrations and first indirect aerosol effect over Moscow in the spring period during the COVID-19 lockdown and during the springs in 2018–2019 according to the observations and numerical experiments. Based on MODIS retrieval, the average cloud number concentration over Moscow was about 200–300 cm$^{-3}$ in April and May 2018–2020. However, during the lockdown in spring 2020, for the periods of northern air advection, we detected a reduction in droplet number concentration of about 14–16% (40–50 cm$^{-3}$) compared to the values in the same conditions in 2018–2019. The reduction in $N_d$ was accomplished by an increase in the effective radius of droplets by an average of 8% and a decrease in the optical thickness of clouds by an average of 5% according to MODIS observations. A decrease in cloud condensation nuclei number concentration in the spring of 2020 was found in both datasets with a resolution of 1 km and 5 km. The satellite-based $N_{CCN}$ retrievals have a wide range of uncertainties due to the cloud heterogeneity, which provides a complex three-dimensional structure of radiative transfer along the cloud edges, etc. [47]. We tried to reduce the regular errors of $N_{CCN}$ retrievals, but even with careful evaluation, the uncertainty in retrieved $N_{CCN}$ is high [38,47]. Another method for qualitative $N_{CCN}$ assessment can be achieved using alternative sources of $N_{CCN}$ data (for example, observations of cloud radar) [24].

The decrease in CCN number concentration by about 14–16% is lower than the received reduction in PM$_{10}$ near the ground by about 30–40% in Moscow [8]. We attribute this discrepancy to complex processes in the planetary boundary layer of the urban area, the influence of atmospheric stratification. However, the effect of decreasing the pollution during the lockdown period should be more pronounced at ground level, located closer to the emission source, which is mainly transport in our case [62]. We plan to add the analysis of vertical aerosol structure in Moscow and its influence on CCN.

We showed good agreement (2%, $11 \pm 27$ W/m$^2$) of CERES global irradiance retrievals with ground-based observations over the Meteorological Observatory of Moscow State University with r = 0.84 in cloudy conditions, which provides a tool for reliable spatial comparisons of model and CERES satellite data over the whole area.

Using the cloud microphysical retrievals from satellite observations, we carried out numerical experiments with the COSMO model. In the process of comparing the observed and simulated cloud characteristics and radiative fluxes, we obtained a significant underestimation of liquid water path and cloud optical thickness and overestimation of global irradiance at ground. To eliminate the possible impact of model errors on the simulated aerosol indirect effects, we considered only relative changes. According to COSMO model simulations, we show that due to a $N_{CCN}$ decrease by 50 cm$^{-3}$, the liquid water path decreases, on average, by 4%. Sensitivity of solar irradiance at ground to the changes in $N_{CNN}$ comprises a 5–9 W/m$^2$ (or 9–11%) increase to the reduction in cloud droplet concentration by $N_{CCN}$ = 50 cm$^{-3}$ in overcast cloud conditions with LWP = 200–400 g/m$^2$ at a solar height of 40°. These estimates are in agreement with the idealized numerical experiments of warm stratus shown in [61].

**Author Contributions:** J.S. and N.C. analyzed the observation data. J.S. and M.S. designed, carried out and evaluated numerical experiments. All the authors wrote, edited and finalized manuscript. All authors have read and agreed to the published version of the manuscript.

**Funding:** This research was funded by the Government of Russia (grant number 075-15-2021-574) and fulfilled at the Lomonosov Moscow State University under the Development Program of the MSU Interdisciplinary Scientific and Educational School "Future Planet and Global Environmental Change". The implementation and sensitivity studies of two-moment microphysics scheme in COSMO-Ru was applied within Roshydromet Research Work number AAAA-A20-120021490079-3.

**Institutional Review Board Statement:** Not applicable.

**Informed Consent Statement:** Not applicable.

**Data Availability Statement:** Not applicable.

**Acknowledgments:** The study of CLOUDRAD scheme is provided within CAIIR priority project (http://cosmo-model.org/content/tasks/priorityProjects/caiir, accessed on 11 August 2022).

**Conflicts of Interest:** The authors declare no conflict of interest.

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
