# Peer review of "Impact of Cloud Condensation Nuclei Reduction on Cloud Characteristics and Solar Radiation during COVID-19 Lockdown 2020 in Moscow"

_atmosphere, doi:10.3390/atmos13101710_

Round 1
Reviewer 1 Report
This study focused on the impact of CCN reduction during the pandemic Lock-down period on the characteristics of cloud and solar radiation in Moscow. The authors used MODIS data during the spring season from 2018-2020. During lockdown period a reduction in droplet number concentration was observed compared to previous years 2018-2019. In connection, an increase in droplet effective radius and a decrease in the optical thickness of cloud was also observed.
The authors carefully selected data and did a lot of analysis. The anti-Twomey effect is observed and estimated the variation in the global radiation due to the change in the cloud microphysical properties using the COSMO model. Very few studies reported variation in cloud properties during the Lockdown period. The manuscript is well written, however, the authors should carefully go through the manuscript for editorial corrections. Some doubts and corrections are given below. The manuscript should go through a minor review before it is publishable.
Page 2, Line 57: Change Nd to “Nd”
Page 5, Line 199: Nd values in the 2018 were the lowest among the presented years. What was the average Nd value during 2018, have you checked why the concentration was lower during 2018?
Page 9, Line 330-333: A less pronounced decrease of the droplet number conc…. The statement seems to be speculative, please rewrite or provide any reference?
Page 11, Line 415: “This decrease was accompanied by an increase in the effective radius of droplets…” may rewrite as “The reduction in Nd was accomplished by an increase in the effective radius of droplets…”
Page 11, Line 411: The first line may rewrite as “According to the MODIS retreival, the average clod droplet number concentration over Moscow region was about 200-300 cm-3 in April and May 2018-2020.”
Page 11, Line 413 “detected the” may rewrite as “detected a reduction”
Author Response
Dear reviewer, the responses are given below. You can find the MS Word file with the "track changes" function in attachments.
Point 1: Page 2, Line 57: Change Nd to “Nd”
Response: The error has been fixed (line 57).
Point 2: Page 5, Line 199: Nd values in the 2018 were the lowest among the presented years. What was the average Nd value during 2018, have you checked why the concentration was lower during 2018?
Response: We checked the meteorological conditions during the spring of 2018-2020. For a better understanding, we’ve added information about the Nd and air temperature to the text (lines 222-227).
“On average, the lockdown spring of 2020 is not distinguished by particularly low concentrations compared to other years. This is because Nd depends more on the synoptic conditions and on the prevalence of specific situations during the period. The median of daily Nd was 184 cm-3, 247 cm-3 and 194 cm-3 for the 2018, 2019 and 2020 respectively. Nd values in 2018 were the lowest among the presented years due to the colder spring conditions. The average spring air temperature at 2 m was 6ºÐ¡ in 2018, 8 ºC in 2019 and 7 ºC in 2020.”
Point 3: Page 9, Line 330-333: A less pronounced decrease of the droplet number conc…. The statement seems to be speculative, please rewrite or provide any reference?
Response: We agree with the remark. We tried to explain this phrase in more detail (lines 356-361).
“It can be assumed that a less pronounced decrease of the Nd compared with the decrease in surface PM10 concentrations of about 30-40% [8, 74] can be explained by the location of the major-changed aerosol sources of emission near the ground and characteristics of planetary boundary layer over Moscow. The study [76] noticed the complex structure of aerosol profile and high dependence of aerosol on structure of planetary boundary layer.”
Point 4: Page 11, Line 415: “This decrease was accompanied by an increase in the effective radius of droplets…” may rewrite as “The reduction in Nd was accomplished by an increase in the effective radius of droplets…”
Response: The phrase has been changed in accordance with the proposed version (lines 442-443).
“The reduction in Nd was accomplished by an increase in the effective radius of droplets by an average of 8% and a decrease in the optical thickness of clouds by an average of 5% according to MODIS observations.”
Point 5: Page 11, Line 411: The first line may rewrite as “According to the MODIS retreival, the average clod droplet number concentration over Moscow region was about 200-300 cm-3 in April and May 2018-2020.”
Response: The phrase has been changed in accordance with the proposed version (lines 438-439).
“Based on MODIS retrieval, the average cloud number concentration over Moscow was about 200-300 cm-3 in April and May 2018-2020.”
Point 6: Page 11, Line 413 “detected the” may rewrite as “detected a reduction”
Response: The error has been fixed (line 440).
“However, during the lockdown in spring 2020 for the periods of northern air advection we detected a reduction of droplet number concentration..”

Reviewer 2 Report
The article is devoted to the study of the influence of aerosol concentration (CCN) on the concentration and size of cloud drops during the covid lockdown in Moscow, as well as changes in solar radion. Such studies are of interest for atmospheric physics. In general, the article is suitable for publication in the journal "Atmosphere".
The reviewer recommends paying attention to the "Discussion and Conclusions" section, it looks very short and sparse. I would like the authors to emphasize the novelty of the study, as well as to present their results in more detail.
Figures 2-3 looks unbalanced and needs review.
Author Response
Dear reviewer, the responses are given below. You can find the MS Word file with the "track changes" function in attachments.
Point 1: The reviewer recommends paying attention to the "Discussion and Conclusions" section, it looks very short and sparse. I would like the authors to emphasize the novelty of the study, as well as to present their results in more detail.
Response: We agree with remarks. We’ve expanded the Section 4 and considered the results in more detail.
Point 2: Figures 2-3 looks unbalanced and needs review.
Response: Now these are Figures 6 and 7. Taking into account the number of tables and plots, we’ve tried to use the manuscript’s space in the beneficial way and, at the same time, to group the figures logically and clearly. We’ve presented to the reader only the most important results in the figures form.

Reviewer 3 Report
Review on “Impact of cloud condensation nuclei reduction on cloud characteristics and solar radiation during COVID-19 lockdown-2020 in Moscow”
General comments:
In this paper, the authors analyzed the impact of CCN reduction on cloud features and solar radiation during COVID-19 lockdown-2020 in Moscow using MODIS satellite, numerical simulation, and observation data in the case of the northern clear air advection. The manuscript was organized well, but there are still some to be corrected before publication. Therefore, I recommend that this paper can be accepted after major revision.
Comments:
1. Lines 74 and 78: Authors need to describe the method to get the information such as cloud optical thickness and droplets’ effective radius for better understanding. Are those elements measured from MODIS?
2. Lines 93-94 and Line 108: I would like to make sure if MODIS NIR has 5 km resolution and MODIS cloud amount data has 1 km resolution. The authors had better describe the resolution of the data clearly.
3. Lines 133-139: I would like to recommend that the authors would provide some images or data obtained from EMEP, TNO-MACC II, and HYSPLIT to make sure the remote sources of emissions did not affect the Nd difference.
4. Lines 176-177: Are there any reasons the authors did not set the 1 km resolution of the model simulation? Because MODIS has 1 km and 5 km resolution.
5. Section 2: The authors had better show the location map of the study area to know the region where the observation is and where the model target area is. And I would like to recommend that the authors would give the flowchart of the study to give readers a better understanding.
6. Section 4: There is no discussion and conclusions in this Section. The authors had better describe the discussion and conclusions not only the summary.
Author Response
Dear reviewer, the responses are given below. You can find the MS Word file with the "track changes" function in attachments.
Point 1: Lines 74 and 78: Authors need to describe the method to get the information such as cloud optical thickness and droplets’ effective radius for better understanding. Are those elements measured from MODIS?
Response: All elements were obtained from MODIS. We described the data more precisely at the beginning of Section 2 (lines 65-80). We’ve also added a table (Table 1) for a better understanding of MODIS data
Point 2: Lines 93-94 and Line 108: I would like to make sure if MODIS NIR has 5 km resolution and MODIS cloud amount data has 1 km resolution. The authors had better describe the resolution of the data clearly.
Response: We’ve added a Table (Table 1) with a brief description of the MODIS data
Point 3: Lines 133-139: I would like to recommend that the authors would provide some images or data obtained from EMEP, TNO-MACC II, and HYSPLIT to make sure the remote sources of emissions did not affect the Nd difference.
Response: Maps with the annual PM2.5 and PM10 anthropogenic emissions from the TNO-MACC II and EMEP databases were added (Figure 2). We also showed the HYSPLIT backward trajectories for selected days for numerical experiments (NWP days).
Point 4: Lines 176-177: Are there any reasons the authors did not set the 1 km resolution of the model simulation? Because MODIS has 1 km and 5 km resolution.
Response: The grid step 1.1 km of COSMO over Moscow region is associated with the data of initial and boundary conditions. We used the initial conditions according to the ICON model (global version) with 13.2 km grid step. Based on the ICON data, we carried out a coherent simulations using COSMO with grid steps 6.6 km (the grid step is twice less than 13.2 km), 2.2 km (three times less than 6.6 km) and 1.1 km (twice less than 2.2 km). Thus, we followed the ICON-model grid refinement rules. At the same time, the simulation results have a close grid step to MODIS data.
Point 5: Section 2: The authors had better show the location map of the study area to know the region where the observation is and where the model target area is. And I would like to recommend that the authors would give the flowchart of the study to give readers a better understanding.
Response: We agree with this remark. We’ve added a flowchart (Figure 1) and a target research area (Figure 3) with some description.
Point 6: Section 4: There is no discussion and conclusions in this Section. The authors had better describe the discussion and conclusions not only the summary.
Response: We agree with this remark. We’ve expanded the Section 4 and considered the results in more detail.

Round 2
Reviewer 3 Report
I think that the authors modified the manuscript according to the comments. Therefore, I would like to recommend accept as in its present form.